# The Potential Role of Serum and Exhaled Breath Condensate miRNAs in Diagnosis and Predicting Exacerbations in Pediatric Asthma

**DOI:** 10.3390/biomedicines11030763

**Published:** 2023-03-02

**Authors:** Natalia Kierbiedź-Guzik, Barbara Sozańska

**Affiliations:** 114th Paediatric Ward—Pulmonology and Allergology, J. Gromkowski Provincial Specialist Hospital, ul. Koszarowa 5, 51-149 Wrocław, Poland; 21st Department and Clinic of Paediatrics, Allergology and Cardiology Wrocław Medical University, ul. Chałubińskiego 2a, 50-368 Wrocław, Poland

**Keywords:** miRNA, asthma, children, serum, exhaled breath condensate, diagnosis, exacerbation, biomarkers, factors

## Abstract

Asthma is the most common chronic disease of the respiratory system in children and the number of new cases is constantly increasing. It is characterized by dyspnea, wheezing, tightness in the chest, or coughing. Due to diagnostic difficulties, disease monitoring, and the selection of safe and effective drugs, it has been shown that among the youngest patients, miRNAs fulfilling the above roles can be successfully used in common clinical practice. These biomolecules, by regulating the expression of the body’s genes, influence various biological processes underlying the pathogenesis of asthma, such as the inflammatory process, remodeling, and intensification of airway obstruction. They can be detected in blood serum and in exhaled breath condensate (EBC). Among children, common factors responsible for the onset or exacerbation of asthma, such as infections, allergens, air pollution, or tobacco smoke present in the home environment, cause a change the concentration of miRNAs in the body. This is related to their significant impact on the modulation of the disease process. In the following paper, we review the latest knowledge on miRNAs and their use, especially as diagnostic markers in assessing asthma exacerbation, with particular emphasis on the pediatric population.

## 1. Introduction

Modern medicine is constantly developing, and new, minimally invasive methods are sought to facilitate accurate diagnosis and control the course of the disease. Undoubtedly, a breakthrough event was the recognition of microRNA molecules (miRNAs) which are involved in many biological processes in cells. These biomolecules can be found in blood serum, tissues, and exhaled breath condensate (EBC). They were discovered more than 30 years ago, but only recently have they become a special object of interest among scientists [1,2,3]. By changing their concentration in the blood serum and knowing the target site of action, they may serve as an extremely useful diagnostic and therapeutic tool allowing us to monitor the course of the disease and detect its exacerbations. Based on a large number of studies, it has been shown that they play a key role in the pathogenesis of allergic diseases (including asthma) by affecting the change in gene expression and modulation of inflammatory processes [4]. Thanks to their minimal invasiveness and the ease of obtaining them, miRNAs are becoming an extremely useful tool in the hands of physicians, especially among the pediatric population. The correct diagnosis of early childhood asthma among this group is often a major clinical problem [5]. Spirometry, i.e., a functional test of the respiratory system, is the gold standard for diagnosing asthma but it requires the cooperation of the patient which is only possible when the child is five to six years old. Due to the still small amount of data on the use of miRNAs among the pediatric population, there is a further need for research to assess their clinical utility in a group of young patients at high risk of an adverse course of the disease.

MiRNAs are small, non-coding RNA molecules, which usually consist of 18–25 nucleotides responsible for regulating gene expression at the translation level and affecting messenger RNA (mRNA) [1]. By binding to the 3′-UTR region, mRNAs lead to the inhibition of translation or degradation in this molecule. These biomarkers act by inhibiting the production of relevant proteins; therefore, it is assumed that the pro-inflammatory effects are probably due to indirect mechanisms [3,6,7]. Approximately 50% of the human genome is regulated by these molecules at the translation stage, which is why they have a significant impact on ensuring homeostasis in the body and coordination of the cell cycle, differentiation, apoptosis, and other physiological functions of cells [8]. Through detailed characterization of the miRNA profile in a given disease entity and correlation of this profile with the appropriate genes, a deeper understanding of the pathophysiological processes will be possible [1,4]. These biomolecules are characterized by high stability and tissue specificity, which emphasizes their additional advantages as new diagnostic markers [8]. MiRNAs have also been shown to be a promising target for potential therapeutic intervention [6,7]. We can divide them into intracellular and extracellular molecules. Thanks to the so-called exosomes, they are transported and transferred between different tissues and cells within the body [3]. Extensively conducted studies have shown a difference in the expression of miRNAs between asthmatics and healthy people. In some of them, the detailed molecular mechanism responsible for the biological processes taking place in cells has already been identified, but most of them are still undiscovered [3,6,7].

Asthma is a chronic inflammatory disease of the bronchial tree characterized by recurrent symptoms, such as cough, shortness of breath, wheezing, and chest tightness, that change in intensity over time. It is the most common chronic disease of the lower respiratory tract in children, usually diagnosed before the age of five (one to two years 34%, less than one year 32%). This may suggest over-diagnosis, because children often cough and wheeze with colds and chest infections, but this is not necessarily asthma [9]. In older children, lung function tests can be used to aid the diagnosis [10]. MiRNA in asthma, through indirect mechanisms, is responsible for the severity of inflammation, hyperresponsiveness, and remodeling of the airways, as well as resistance to standard therapy with inhaled steroids [6]. Moreover, allergens, infections, and air pollution may cause intensification and exacerbation of asthma (especially among children), change the concentration of miRNAs in the serum, and modulate the course of the disease process. The following publication, based on a literature review, aims to introduce miRNAs as new diagnostic markers that allow us to monitor the course of the disease, as well as to recognize and predict its exacerbation, with particular emphasis on the pediatric population [5]. This potential role of miRNAs in pediatric asthma we presented in Figure 1.

## 2. The Diagnostic Role of miRNAs

Due to diagnostic difficulties, delays in initiating appropriate treatment and the need to deal with complications of untreated diseases, asthma is a resource-consuming condition in the healthcare sector. Finding and implementing into clinical practice non-invasive, fast, and sensitive diagnostic methods that can undoubtedly be based on miRNA molecules would largely solve this growing problem. Research has been conducted on these biomolecules that could play a role in this process. Increased expression of miRNA-221 and miRNA-485-3p has been shown in the pediatric asthma population. The first molecule is associated with increased adhesion and migration of mast cells and cytokine production after the body comes into contact with the antigen. In addition, these biomolecules reduce the concentration of the Spred-2 protein (Sprouty-related EVH1 domain-containing 2), contributing to increased cell proliferation and the eosinophilic inflammatory response in the airways through IL-5 [7,11]. MiRNA-21 levels are higher in children with confirmed asthma and positively correlated with disease severity and serum and sputum eosinophilia in these patients. This molecule acts on the Smad-7 gene and inhibits the production of a protein that is an inhibitor of the TGFβ1/Smad pathway. The consequence of this is the excessive synthesis of collagen, α-smooth muscle actin, proliferation and differentiation of fibroblasts, and deposition of the extracellular matrix. This results in remodeling, fibrosis of the airways, and increased airway obstruction [12]. In a study conducted by M. Atashbasteh et al. among patients with severe asthma, increased expression of miRNA-125 was demonstrated, as well as decreased expression of miR-124, miR-130a, and miR-133b. The level of the first of these was correlated with the concentration of CRP and IgE in the serum. These molecules are involved in the pathway of phosphorylation of sphingosine to sphingosine-1 phosphate (S1P) by sphingosine kinase type 1 (SphK1). The effect of S1P on cells is possible by affecting receptors, e.g., S1PR and S2PR. The former is present in large amounts on the surface of lymphocytes and is involved in the process of their maturation (in particular, release from lymphatic organs). In addition to affecting the immune system cell population, S1P also participates in the activation of molecules that regulate inflammation, proliferation, and cell death: NF-ĸB (nuclear factor kappa B) and STAT3 (signal transducer and activator of transcription 3). For this reason, the significant role of this pathway in the development of autoimmune diseases is emphasized. The increased concentration of this compound in the airways of patients with asthma is responsible for the increased hyperreactivity of the smooth muscle cells of the bronchial tree and the increased inflammatory process. MiRNA-124 is associated with sphingosine kinase 1 (SphK1), which is responsible for the formation of S1P, miRNA-130a with an effect on the S1PR2 receptor (it was proven that the reduction in its expression is associated with an increase in the concentration of inflammatory mediators, including TNF-α and increased expression of inflammatory genes). MiRNA-133b, on the other hand, plays a significant role in controlling the level of sphingosine-1-phosphate receptor protein [13].

An increased blood concentration of some miRNAs in the pediatric population suffering from asthma, e.g., miRNA-3162-3p, miRNA-1260a, miRNA-let-7c-5p, and miRNA-494, has become the subject of further research interest in understanding their role in the pathogenesis of allergic diseases. Particular attention was paid to the miRNA-3162-3p increase in serum in a study of an allergic asthma model conducted in albumin-sensitized mice. Its relationship with β-catenin, which is responsible for the reconstruction of the respiratory tract, and its inhibitory effect on the expression of the gene encoding this protein was proven. The experimental use of antagomir (anti-miRNA) resulted in a reduction in airway hyperreactivity, inflammation, and an increase in B-catenin levels. Thus, signaling involving this molecule seems to be crucial in models of asthma provoked by allergens [14,15].

In childhood the model of allergic asthma dominates, with an increased Th2 cell response and eosinophilic inflammation of the airways. Eosinophils play a key role in the development and maintenance of the inflammatory process. It was shown that by their effect on smooth muscle cells (ASMC) in laboratory conditions, they promote proliferation and hypertrophy. This only occurs in the environment of eosinophilic cells obtained from the blood of sick people. These processes include, for example, TGF-β1 and WNT-5 genes [16]. Eosinophils are also a source of miRNAs which can be transferred between the cells of the body through exosomes, becoming a potential regulator of gene expression [17].

In a study evaluating differences in the expression of miRNAs in eosinophils in patients with the allergic march and healthy subjects, 18 biomarkers were isolated that were associated with the expression of genes involved in cellular regulation, immune response, angiogenesis, and smooth muscle cell proliferation. This highlights the role of miRNAs, which may play a role in the atopic march and the occurrence of many allergic diseases in one patient. Attention was paid to miRNA-590, which was then also shown to be down-expressed in previous studies. Its target is the CITED-2 gene, which is responsible for the regulation of the TGF-beta pathway and the proliferation of airway smooth muscle cells [18,19]. Another study demonstrated a higher concentration of miRNA-144-3p in severe disease and its positive correlation with blood eosinophilia. This biomolecule affects genes responsible for the inflammatory process and remodeling of the airways, e.g., GATA3, STAT6, SOCS5, RHOA, NR3C1, and PTEN, which makes it a potential diagnostic marker for severe asthma [20]. In non-allergic asthma, miRNA-629-3p, miRNA-223-3p, and miRNA-142-3p were significantly elevated in patients with severe symptoms and associated with increased inflammation mediated by neutrophils. In particular, the miRNA-629-3p molecule was responsible for the increased production of IL-8, which stimulates the migration of neutrophils, monocytes, and T cells and the adhesion of neutrophils to the endothelium [21]. Individual miRNAs also affect the expression of interleukins. Involvement in inflammatory processes, e.g., miRNA-1248 (increased concentration in asthmatics) stimulates the synthesis of IL-5, responsible for survival, growth, differentiation, and recruitment of eosinophils. MiRNA-181b-5p, on the other hand, showing reduced expression in people with asthma, is characterized by targeting the SPP1 molecule, i.e., phosphoprotein 1, otherwise known as osteopontin. It is a component of the extracellular matrix that is involved in the migration of eosinophils to the respiratory tract and the intensification of IL-13-induced expression of IL-1 and CCL 11 (eosinophil chemotactic protein) in bronchial epithelial cells [7,22]. MiR-146a and miR-106b, in turn, are upregulated in pediatric asthma patients and lead to increased production of IL-5 and IL-13, which stimulate inflammatory cell recruitment, epithelial cell hyperplasia, smooth muscle hyperplasia, goblet cell metaplasia, and extracellular matrix deposition in the respiratory tract [23].

## 3. Asthma Exacerbation and miRNAs

MiRNA molecules perform various functions in the body. In addition to the diagnostic role described above, they can be used to predict exacerbation of the disease with greater accuracy. The risk of exacerbation of the disease increases with the lack of use of inhaled corticosteroids, lack of adequate disease control, spirometric indices indicating deepening of obstruction, signs of eosinophilic inflammation, or exposure to allergens or irritants. In a group of children diagnosed with asthma and using anti-inflammatory treatment (inhaled steroids) for a period of 12 months, 12 miRNA molecules associated with exacerbations during observation were isolated. In addition, each doubling of the concentration of these molecules was associated with an increase in the risk of exacerbations by 25–67%, respectively. MiRNA-146b-5p, miRNA-206, and miRNA-720 were identified, which, in combination with clinical symptoms, enabled a better prediction of exacerbation of the disease in patients with asthma using inhaled corticosteroids, compared to a model based only on a single component. Involvement of these miRNAs in pathways involved in the pathogenesis of asthma was discovered, e.g., NF-kβ and GSK3 (glycogen synthase kinase-3)/AKT, which are responsible for remodeling the airways and deepening the inflammatory process [2].

MiRNAs are biomolecules that also play a key role in other obstructive respiratory diseases, including chronic obstructive pulmonary disease (COPD). Researchers compared circulating serum miRNAs associated with exacerbation of the disease in the pediatric population diagnosed with asthma and adults diagnosed with COPD. A total of 20 miRNA molecules were associated with worsening of symptoms in children and 5 of them (451b; 7-5p; 532-3p; 296-5p, and 766-3p) in the adult population with severe chronic obstructive pulmonary disease. Participation of these molecules in the signaling pathways MAPK (mitogen activated protein kinases) and PI3K-Akt, which are responsible for the increased response of both eosinophils and neutrophils, the production of immunoglobulin IgE, and the activation of tumor necrosis factor alpha (TNF-a), was proven. This leads to excessive production of interleukins, cytokines, and intensification of inflammation [24]. In addition, the MAPK pathway is activated by tobacco smoke, indicating its important role in the pathogenesis of COPD and neutrophilic asthma. Thanks to the analysis of the concentrations of appropriate miRNAs, it is also possible to identify the pathomechanism that may lead to exacerbation of symptoms among children with asthma who are exposed to tobacco smoke in the home environment. This is undoubtedly a factor that stimulates an increased immune response and affects the disease symptom severity. However, anti-inflammatory molecules have been identified that are also related to the concentration of the relevant miRNAs. One of them is annexin (ANXA1), also known as lipocortin I. It is a protein with a significant content in the secretions of the respiratory tract. It was shown that an increase in ANXA1 concentration was associated with the decreased concentration of miRNA-196-a2. It is also interesting that in subjects with a moderate disease severity, the level of ANXA 1 was higher than in those with severe asthma. In addition, miRNA-196-2a is responsible for the production of key interleukins involved in the pathogenesis of asthma, IL-5 and IL-13, whose role is described above [25,26,27]. Many factors can exacerbate the course of asthma, such as tobacco smoke, infections (mainly viral), allergens, irritants (e.g., aerosols or household cleaners, paint fumes, other occupational exposures), physical exertion, air pollution, medications, and foods. Respiratory infections caused by viruses, including RSV, influenza, and rhinoviruses, are of particular importance among the group of the youngest patients. When these pathogens infect human bronchial epithelial cells (HBECs), it was demonstrated that the NF-κB pathway and interferon signaling are activated to stimulate cellular responses, reduce viral replication, and avoid tissue damage. It was suspected that the HBEC cells of asthmatics impair the above-described processes and, consequently, exacerbate symptoms. Understanding miRNAs targeting the NF-κB and interferon pathway will uncover modulators of cellular responses and prevent the development of adverse events [28]. MiRNA-146a and miRNA-146b (miRNA-146a/b) are anti-inflammatory molecules more heavily produced in response to rhinovirus (RV) infection, targeting the NF-κB pathway specifically. In experimental animal models of allergic asthma, deprivation of HBEC cells of these miRNAs led to an increased inflammatory process involving Th1, Th17 lymphocytes with a reduced participation of Th2 lymphocytes. These molecules are therefore responsible for alleviating RV-induced allergic airway inflammation and represent a potential future therapeutic target [29]. Influenza virus is another important infectious agent responsible for exacerbating asthma symptoms. After infection of bronchial epithelial cells with H1N1 influenza virus in the laboratory, miRNA-22 growth was demonstrated only in a sample obtained from healthy individuals. The molecule blocks the CD147 receptor which is a transmembrane glycoprotein involved in the invasion of viral and bacterial infections. This molecule also participates in the remodeling of the respiratory tract through the increased synthesis of matrix metalloproteinases (MMPs). Therefore, asthma patients with low levels of miRNA-22 due to the inability to block CD147 lose one of the important defense barriers against viral infection, becoming more susceptible to infection. In the era of the COVID-19 pandemic, it was shown that the CD147 receptor is also responsible for the penetration of the SARS-CoV-2 virus into the cell. This proves the role of miRNA-22 in the defense process against a wider spectrum of viral infections than just influenza [30].

Unfortunately, not only infectious factors play a leading role in the pathogenesis of asthma in children, but also increasing air pollution and high concentrations of harmful substances in the environment. After measuring the content of PM 2.5 particles in the composition of the air at home, a significant correlation was shown between their concentration and the incidence of asthma. Moreover, greater exposure resulted in an increase in serum concentrations of miRNA-155, which is responsible for the enhancement of the type 2 immune response [31]. The relationship of this biomolecule was also confirmed with exposure to tobacco smoke, which is a special type of air pollution due to its local occurrence and relative ease of elimination from inhaled air compared to other factors. MiRNA-155-5p, miRNA-21-3p, and miRNA-18a-5 are a set of molecules that are overexpressed in the blood of mice exposed to cigarette smoke in utero, in which asthma was later induced with albumin. Additionally, their concentrations were positively correlated with proasthmatic Th2 cytokine levels in bronchoalveolar lavage fluid (BALF) samples [32]. Another aspect that was emphasized was the seasonal variability of miRNAs and its impact on the severity of the allergic process and asthma symptoms in the pediatric population, especially in the spring. It is at this time of the year patients are at risk of exacerbating asthmatic symptoms, especially those with hay fever. Of the 26 miRNAs studied, which were associated with a specific season and with the allergen causing the symptoms of this period, two miRNAs-328-3p and let-7d-3p were isolated. A decrease in the concentration of let-7d-3p was observed in spring and among those allergic to mulberry (spring blooming), while it increased after the allergen immunotherapy process (performed in patients allergic to wasp venom). The protective role of this molecule was proven, which is most likely achieved by reducing the concentration of IL-13. Its protective effect in relation to asthmatic patients is also supported by its abundant presence in the lung tissue. The role of miRNA-328-3p is more complex. It was observed that its concentration decreases in autumn. On the other hand, an increase was noted in patients with a concomitant allergy to aspergillus, which resulted in an exacerbation of symptoms. MiRNA-328-3p was shown to be involved in wound healing of the bronchial epithelium, but also facilitates the spread of bacterial infection in the lungs [33].

MiRNAs may participate in the process of airway remodeling, leading to airway obstruction and affecting airway function parameters. MiRNAs involved in a number of pathophysiological processes affect smooth muscle cells, the epithelium, and goblet cells and intensify the inflammatory process, causing narrowing of the bronchial lumen [34].

The relationship between 22 miRNAs and lung function parameters in the pediatric population was demonstrated. An example may be the following molecules: miRNA-186-5p, which participate in the release of acetylcholine and modulation of airway tone through the cholinergic pathway [35,36]; miRNA-203, which is associated with an increase in IgE concentration and intensifies the inflammatory process in the airways, leading to their obstruction [35,37]; and miRNA-26, which is released by bronchial smooth muscles after their physical stretching, causing cell hypertrophy. Over the course of many years of observation, it has also been proven that miRNA-145-5p is associated with an early decrease in FEV1 in children with asthma, leading to the development of COPD. It is also responsible for the increased proliferation of airway smooth muscle cells [38].

The results of the studies on miRNAs in pediatric asthma are summarized in Table 1.

## 4. Exhaled Breath Condensate

Examination of the exhaled breath condensate (EBC) is a new, non-invasive method that allows us to assess the inflammatory process within the respiratory tract and may be successfully used in children. The material is gathered by calm breathing for 10–15 min, and then cooling and accumulating the air exhaled into the capacitor. Thanks to the obtained condensate, it is possible to measure inflammatory markers in the collected material, as well as miRNA. In this way, a potentially useful, non-invasive technique was obtained for diagnosing and controlling the course of the disease and assessing the effectiveness of asthma treatment. The presence of miRNAs in EBC was confirmed for the first time in 2013. The difference in the expression of 11 miRNAs between asthmatics and healthy people was proven, as well as the greater stability of miRNA molecules in EBC compared to the material obtained from blood serum due to the difference in the number of miRNAs enclosed in exosomes [41,42].

It was found that the concentration of miRNA contained in EBC is related to the functional parameters of the respiratory tract. In the first study conducted among the pediatric population with diagnosed asthma in which the exhaled breath condensate was analyzed, the influence of a number of miRNAs (e.g., miRNA-155, miR-126-3p, miR-133a-3p, miR-145-5p, 3p) on lung function parameters and reversibility of airway obstruction was proven [39]. In subsequent studies, the relationship between miRNA-570-3p in EBC and spirometry results was also found through an inverse relationship between the expression of this molecule and FEV1 values. A similar relationship was obtained after examining miRNA-1248. With regard to the intensity of the inflammatory process, it was found that miRNA-570-3p affects the diverse expression of many cytokines, chemokines, and the HuR protein (it binds RNA and regulates post-transcriptional processes). Thus, this biomolecule becomes a potential regulator of inflammation in asthma [43]. In addition, attention was paid to miRNA-423, the relationship of which was proven with obesity. Obesity is known to be one of the factors predisposing a higher risk of developing asthma. Since asthma and obesity are characterized by chronic inflammation, it is likely that miRNAs may be misregulated in these diseases by modulating the immune cells found in EBCs. Interestingly, an unobvious relationship between the increase in the amount of fatty acids in the food consumed and the increase in the level of miRNA-133a-3p in the EBC affecting the inflammatory response in the condensate of exhaled air among children diagnosed with asthma was also noted [40].

The limitation of our work is the fact that some studies have been based on relatively small pediatric populations or studies with no control group. To present the trends and potential perspectives for future studies, we have presented studies conducted on adult populations or animal models in case of a lack of data on children. 

## 5. Conclusions

In modern medicine we strive for an individual approach to each patient. This strategy is mainly based on clinical phenotyping, where biomarkers play an important role. Additionally, in childhood diseases, especially in asthma, problems with diagnostics and assessment of treatment efficacy often occur due to a lack of cooperation with young patient. Thus, many molecules, metabolites, and proteins remain the subject of research to find a simple, useful biomarker that may solve the abovementioned problems. The potential relevance of molecules, such as (in blood) eosinophil cationic protein, periostin, lipoxins, chitinases, YKL-40, (in exhaled breath) fractional exhaled nitric oxide, volatile organic compounds, evaluation of exhaled breath temperature, (in urine) bromotyrosine, metabolites of eicosanoids, eosinophil-derived neurotoxin in diagnostics and management of asthma were evaluated. Unfortunately, these biomarkers presented several limitations, e.g., particles detected in the urine do not directly reflect the inflammatory process in the respiratory tract and the concentration of some molecules changes during the growth of children or in the course of other diseases [44,45,46]. These facts make them useless in clinical practice. Therefore, understanding the importance of new, promising biomarkers (miRNAs) and their role in metabolic pathways in childhood asthma seems to be crucial. Based on the considerations presented in this manuscript, it can be seen that miRNAs, in combination with disease symptoms, lung function tests, and allergy tests, seem to be a useful tool in the construction of predictive models, allowing for the identification of high-risk groups for the adverse course of the disease and increasing the probability of making an accurate diagnosis in an ambiguous case. In order for miRNAs to be introduced into common clinical practice, the methods of their detection should be characterized by high sensitivity and specificity. Based on many studies, it has been shown that they indeed have such potential; however, a lack of data prevents their successful implementation in clinical practice. Therefore, there is a need to conduct new analyses and search for solutions using miRNA molecules in the population of pediatric patients diagnosed or suspected of having bronchial asthma.

## Figures and Tables

**Figure 1 biomedicines-11-00763-f001:**
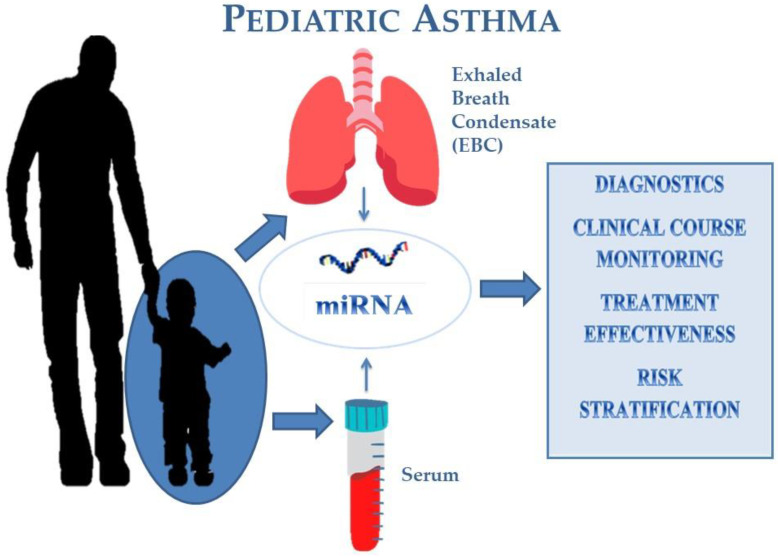
Serum and exhaled breath condensate-isolated miRNA and its potential role in asthma diagnostics and management in the pediatric population.

**Table 1 biomedicines-11-00763-t001:** The relevance of miRNA in asthma based on studies conducted in the pediatric population.

Study	Population	miRNA	Gene	Biological Function
F. Liu et al. [11]	N = 12 (study with control group) Age: 4–6 years SAMPLE: serum	↑ miRNA-221 ↑ miRNA-485-3p	SPRED2	Decreased Spred**-2** protein. Increased cell proliferation and eosinophilic inflammatory response in the airways through IL-5.
Y. Kang et al. [12]	N = 103 (study with control group) STUDY GROUP: ASTHMA Age: 9.3 ± 0.4 years CONTROL GROUP Age 9.6 ± 0.4 years SAMPLE: serum	↑ miRNA-21	Smad7	Decreased Smad7 protein. TGFβ1/Smad pathway. Excessive synthesis of collagen, α-smooth muscle actin, proliferation and differentiation of fibroblasts. Increased airway obstruction.
Y. Wang et al. [14]	N = 54 (study with control group) SAMPLE: serum	↑ miRNA-3162-3p ↑ miRNA-1260a, ↑ miRNA-let-7c-5p ↑ miRNA-494	CTNNB1	Decreased β-catenin. Intensification of airway hyperreactivity and inflammation.
H. Elnady et al. [23]	N = 50 (study with control group) STUDY GROUP: ASTHMA Age: 10.6 ± 0.7 years CONTROL GROUP Age: 11.0 ± 0.8 years SAMPLE: serum	↑ miRNA-146a ↑ miRNA-106b	-	Increased IL-5, IL-13. Stimulate inflammatory cell recruitment, epithelial cell and smooth muscle hyperplasia, goblet cell metaplasia, and extracellular matrix deposition in the respiratory tract.
L. He et al. [18]	N = 170 (study with control group) SAMPLE: serum	↑ miRNA-144-3p	GATA3, STAT6, SOCS5, RHOA, NR3C1 i PTEN	Increased eosinophilia, inflammatory process, and remodeling of the airways.
A. T. Kho et al. [2]	N = 153 (study with control group) STUDY GROUP: ASTHMA EXACERBATION Age: 8.9 ± 2.2 years CONTROL GROUP: NO EXACERBATION Age: 8.9 ± 2.0 years SAMPLE: serum	↑ miRNA-146b-5p ↑ miRNA-206 ↑ miRNA-720	-	Upregulation GSK3 by AKT and downregulation NF-kB pathway. Remodeling the airways and deepening the inflammatory process.
A. Tiwari et al. [24]	N = 351 (study with control group) STUDY GROUP: ASTHMA EXACERBATION Age: 9.0 ± 1.9 years CONTROL GROUP: NO EXACERBATION Age: 9.4 ± 1.8 years SAMPLE: serum	↓ miRNA-451b ↓ miRNA-7-5p ↑ miRNA-532-3p ↑ miRNA-296-5p ↑ miRNA-766-3p)	Many genes involved in the production of more than 20 proteins.	Upregulation MAPK, PI3K-Akt (i.a). Increased eosinophils, neutrophils, IgE immunoglobulin, and the activation of tumor necrosis factor alpha (TNF-a). Excessive production of interleukins and cytokines and intensification of inflammation.
A. A. Ibrahim et al. [25]	N = 100 (study with control group) STUDY GROUP: MILD, MODERATE, SEVERE ASTHMA Age: 8.9 ± 1.3 years CONTROL GROUP Age: 8.2 ± 1.4 years SAMPLE: serum	↑ miRNA-196-2a	ANXA1	Downregukation Annexin (anti-inflammatory factor). Concentration: moderate asthma > severe asthma. Increased inflammatory reaction.
Q. Liu et al. [31]	N = 360 (study with control group) STUDY GROUP: ASTHMA Age: 10.8 ± 3.1 years CONTROL GROUP Age 10.1 ± 2.7 years SAMPLE: serum	↑ miRNA-155	-	Responsible for the enhancement of the type 2 immune response.
A. T. Kho et al. [35]	N = 360 (study without control group) Age: 8.8 ± 2.1 years SAMPLE: serum	↑ miRNA-186-5p ↑ miRNA-203 ↑ miRNA-26	≈50	Activation of cholinergic pathway, intensifies the inflammatory process in the airways leading to their obstruction. Hypertrophy of smooth muscles.
A. Tiwari et al. [33]	N = 398 (study without control group) Age: 5–12 years SAMPLE: serum	↑ let-7d-3p ↑ miRNA-328-3p	-	Decreased serum level of IL-13. Exacerbation of symptoms in patients allergic to aspergilus.
F. C. Mendes et al. [39]	N = 186 (study with control group) STUDY GROUP: ASTHMA Age: 8.7 ± 0.8 years CONTROL GROUP Age: 8.7 ± 0.8 years SAMPLE: exhaled breath condensate	↑ miRNA -155 ↑ miR-126-3p ↑ miR-133a-3p ↑ miR-145-5p, ↑ miRNA-423-3p	E.G RUNX3	GATA-3-upregulation of Th1/Th2 balance. Downregulation of GATA-3. Promotes a lymphocyte Th2 response. Increase in the levels of the IL-13 inflammatory response.
F. C. Mendes et al. [40]	N = 150 (study with control group) Age: 7–12 years SAMPLE: exhaled breath condensate	↑ miR-133a-3p	-	Upregulation of production of IL-13.

Abbreviations: SPRED2—Sprouty-related EVH1 domain-containing 2; Smad—suppressor of mother against decapentaplegic; TGFβ1—transforming growth factor β1; CTNNB1—catenin beta gene; IL—interleukin; GATA3—GATA binding protein three gene; STAT6—signal transducer and activator of transcription six gene; SOCS5—suppressor of cytokine signaling protein five gene; RHOA—Ras homolog family member A gene; NR3C1—nuclear receptor subfamily three group C member 1 gene; PTEN—phosphatase and tensin homolog gene; GSK3—glycogen synthase kinase-3; AKT—protein kinase *B*; NF-kB—nuclear factor kappa-light-chain-enhancer of activated B cells; MAPK—mitogen-activated protein kinases, PI3K—the phosphatidylinositol 3-kinase; Ige—immunoglobulin E; GATA3—GATA binding protein three; Th1—Type 1 T helper; Th2—Type 2 T helper.

## Data Availability

No new data were created or analyzed in this study. Data sharing is not applicable to this article.

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
