# Peer review of "The Potential Role of Serum and Exhaled Breath Condensate miRNAs in Diagnosis and Predicting Exacerbations in Pediatric Asthma"

_biomedicines, 2023, doi:10.3390/biomedicines11030763_

Round 1
Reviewer 1 Report
The manuscript describing the diagnostic role of miRNAs in pediatric asthma is clearly written by there are the following concerns.
1. Cought should be cough.
2. Citations are not as per MDPI format.
3. 5-6 years old Due to-please check
4. lines 49-62- please cite the proper reference for each fact- don't pool the references.
5. before the age of 5 and- please mention years after 5
6. Figure 1 has only title, please add legend.
7. Sections 2, 3, and 4- please include Tables to list all microRNA, genes and function regulated by them, inhibition or activation, type of sample they are diagnosed in (serum, saliva, blood, urine, EBC), pathology involved in asthma.
8. Please discuss noninvasive biomarkers mainly miRNAs for asthma
Please check the language for grammar
Author Response
Dear Sir or Madam,
thank you very much for all comments regarding the article.
The detailed replies to your review are included in the file attached below.
Yours sincerely,
Natalia Kierbiedz-Guzik

Reviewer 2 Report
In this article, the authors reviewed published studies in which miRNA levels in serum or exhaled breath condensate change under conditions like infections, allergens, pollution, etc., and may be of diagnostic value for pediatric asthma.
The use of miRNAs as diagnostic markers is indeed relevant, however the authors report a series of studies without mention study design, population type, sample size, quality of the study, age range, etc., this is important to understand the significance of the results. In addition, the authors should provide strengths and limitations of any of the studies, according to their own professional opinion or experience.
A table or figure showing under- and over-expressed miRNAs, their target genes, and the process they regulate could be included to help the reader appreciate the information in a more integrated manner.
Figure 1 may work as graphical abstract as well (this is just an idea).
The authors conclude that miRNAs have diagnostic potential, however, data are not enough to successfully implement their use the clinic. Therefore the title could be adjusted as follows: “The POTENTIAL role of serum and exhaled breath condensate miRNAs in diagnosis and predicting exacerbations of pediatric asthma”.
Some edition is necessary, I found some typos and minor mistakes in the document.
Author Response
Dear Sir or Madam,
I am grateful for any comments that undoubtedly allowed to improve the quality of the manuscript.
Detailed replies to the comments are included in the file attached below.
Yours sincerely,
Natalia Kierbiedz-Guzik

Round 2
Reviewer 2 Report
In this revised version of the manuscript the authors answered my questions and the effort is appreciated. Some final comments:
Table 1 should be referred somewhere in the text, and I suggest locating the Table towards the end of the manuscript, before the discussion, as an integrative summary of the data.
Some edition of the document is still necessary.
Author Response
Dear Reviewer,
thank you for reviewing our article. Undoubtedly, it contributed to the improvement of its quality. We are grateful for all the valuable comments. Detailed replies to the comments are included in the file attached below.
Yours faithfully,
Natalia Kierbiedź-Guzik
